# Assessment of Actual Weight, Perceived Weight and Desired Weight of Romanian School Children-Opinions and Practices of Children and Their Parents

**DOI:** 10.3390/ijerph19063502

**Published:** 2022-03-16

**Authors:** Anda-Valentina Trandafir, Maria Fraseniuc, Lucia Maria Lotrean

**Affiliations:** Department of Community Medicine, Iuliu Hatieganu University of Medicine and Pharmacy, 400012 Cluj-Napoca, Romania; fraseniuc.maria@yahoo.com (M.F.); llotrean@umfcluj.ro (L.M.L.)

**Keywords:** children, parents, misperceptions, actual weight, bullying

## Abstract

Objective: Children and parental awareness regarding weight is linked to the development and management of overweight and obesity. The aim of this study is to assess the actual weight, perceived weight, and desired weight of Romanian school children. Methods: A cross-sectional study was conducted in 2019 in seven schools from two counties of Romania and included 880 pupils aged between 10 and 15 years old and 665 parents. We administrated confidential questionnaires to the participants, and we measured children’s weight and height during school activities. Results: A total of 61.0% of pupils had normal weight, 7.4% were underweight, and 31.6% were overweight or obese. A total of 66.7% of normal weight children, 56.5% of overweight children, and 40% of underweight children perceived their weight accurately. Regarding parents, a majority correctly appreciated the weight of their normal weight children and only a third appreciated correctly the body weight of their underweight and overweight children. Factors such as body mass index, gender, weight related behaviors, parents’ estimation about their children’s weight, discussions of weight topics inside family, and bullying, cyberbullying and exclusion from groups were associated with misperceptions. Conclusion: The study provides useful information for health education activities targeting both children and their parents regarding appropriate body weight management of children.

## 1. Introduction

The management of body weight represents a public health global concern for governments [1]. According to the World Health Organization (WHO) European region, a third of children are overweight or obese [2]. In the Australian population, 21–25 percent of children between 2 and 18 years old are overweight [3], and in the United States population for 10 years now approximately 1 in 5 children are obese [4]. Childhood obesity tends to continue late in life [5] and is linked to serious complications such as type 2 diabetes, coronary heart disease, asthma, sleep problems, future types of cancer in adulthood, and premature mortality [6,7,8,9,10].

Since excess weight tracks from infancy into adulthood, health interventions should focus on prevention and treatment of obesity at different levels and should start early in life. Policy measures, along with health education in schools, hospitals, or communities represent important strategies in reversing childhood obesity [11]. Among factors that influence the success in fostering healthier weight-related behaviors are subjective aspects such as one’s own perceived weight and the parent’s lack of awareness of their child’s weight status [12,13]. If being overweight is not recognized, children may be less motivated to involve in lifestyle changes. On the contrary, an obese child who admits having weight problems reports more weight loss attempts [14]. Moreover, parents who do not recognize the medical condition of their child may not be prepared to receive interventions or counselling on weight issues [15].

In line with the aforementioned statements, studies showed that teenagers misperceive their weight, considering their size heavier than it really is (overestimation), or smaller than it actually is (underestimation) [16]. Factors associated to misperception such as gender, diet, and maternal attitude were reported [17,18,19,20,21].

Moreover, previous studies showed that parents fail to identify accurately the body weight of their children, with 60% of them underestimating their children’s size [5,22]. If parents misperceive their child’s weight status, they are less likely to encourage healthy lifestyle behaviors in the family [22]. Parents play an important role in children’s choices, thus they may not only influence the development of obesity in their child, but its management, too. An aware parent is more likely to engage in healthier decisions such as a careful diet monitorization, promotion of healthier eating behaviors, seeking treatment, and following a healthcare professional’s recommendations [23,24,25,26]. 

In this context, it is of great interest for parents to understand the health risks of obesity and their role in its prevention, but the biggest challenge would be to correct their misperceptions and apply specific interventions to their children according to their weight [27].

On the other hand, in his research, Eric Robinson concluded that interventions on raising parental awareness regarding the weight status of their child need to be aware of the possible adverse outcomes on the child’s mental health [28]. Literature showed that parental identification on child overweight and obesity is associated with further weight gaining in youngsters [29]. This hypothesis is explained by the fact that heavy weigh children are teased and stigmatized inside their family. Additionally, harsh parenting on weight-control, supported by criticisms and comments, has been associated with emotional disorders in children [28]. 

Moreover, evidence showed that obesity is susceptible to mental health problems, because of weight stigma [30]. Hence, a heavy child who experiences stereotypical portraits, discriminations (e.g., mistreatments and social devaluation), and harsh language towards him is likely to engage in disordered eating [31]. Prior findings suggest that weight stigma is a predictor for future weight gain [32], which can be explained by the fact that it increases the obesogenic stress process and triggers coping mechanisms such as overeating [33]. Individuals may experience, additionally, other forms of weight stigma, such as teasing, bullying, and labelling which can have a significant impact on their well-being, given the high sensitivity of weight issues during adolescence [31]. The “too fat” label, especially when it comes from family, has long-term detrimental effects on health. Considerable studies showed that labelling in childhood led to distorted eating cognitions and behaviors tracked into adulthood, and it was associated with high body mass index (BMI) years later, regardless of the initial weight [31,33]. 

Furthermore, body image is associated with higher rates of body dissatisfaction [34]. This issue is related to children’s seeking an ideal body type established by society [21,35]. Body image has gender characteristics: for girls, body satisfaction is related to weight, facial features, and skin appearance [36], while for boys, it is represented by strength, muscle mass development, and speed [37]. Boys are more likely to engage in physical activities in order to become leaner, fit, and stronger [38], whereas girls restrict their diets in order to become thinner [39]. To lose weight, teenagers may adopt unhealthy solutions such as fasting, vomiting, purging, self-medication with diet pills, smoking cigarettes, food restraints, or extreme diet that eventually may lead to binge eating and bulimic behaviors. [40,41,42]. 

Similar to suicidality, the prevalence of body image dissatisfaction is high in midchildhood [43]. In his publication, Kline outlined that normal weight children who saw themselves heavier reported suicidal thought more often than obese teens who perceived their weight accurately [44]. In addition to suicidal ideation, body image dissatisfaction is comorbid with anxiety, depression, self-injury, and low self-esteem [45].

On the other hand, evidence showed that chronic weight dissatisfaction may affect physical health as well. High weight discrepancy was linked to poor diet, lack of physical activity, tobacco use, and alcohol consumption which consequently led to an increased likelihood of type 2 diabetes, hypertension, and metabolic syndrome later in life [46,47]. On the contrary, people who declared themselves satisfied with their current weight, regardless of BMI, were more likely to adopt healthy lifestyles and had better long-term health outcomes [48]. 

In Romania, national information on the prevalence of overweight and obesity among children is limited. A review of the studies published between 2006 and 2015 showed that 25% of youth aged 6–19 years old were overweight or obese [49]. A recent study showed that prevalence of overweight and obesity among children and adolescents aged 7–18 years old from the western part of the country was 30% and it was highest among pre-pubertal children and boys [50]. 

Starting in the 1990s, Romania has undergone socio-economic changes that might influence the trend of overweight and obesity. To our knowledge, few studies on the Romanian population assessed the association between children’s BMI and weight control behaviors. Additionally, prior research on Romanian children and parents’ perception regarding body weight is scarce and there is no previous evidence about the relationship between children’s weight status and vulnerability to bullying in Romania [27,50]. 

Hence, the present study has four objectives. First, it evaluates the correspondence between anthropometric assessments of Romanian children (actual weight) and their perceptions of their weight (children’s perceived weight), as well as their parents’ perception of their children’s weight (perceived weight by parents), and their concern for body weight management among children (desired weight). Second, the study assesses weight management and lifestyle related behaviors, as well as the prevalence of bullying among participating children and communication between children and parents and between parents and healthcare professionals regarding weight. Third, the factors associated with overweight and obesity are evaluated. Last, but not least, we aim to identify the factors which influence the correct estimation of children’s own weight.

## 2. Materials and Methods

### 2.1. Study Design and Sample

The study was approved by the Ethics Commission of Iuliu Hatieganu Medicine and Pharmacy University, Cluj-Napoca, Romania (134/6.05.2019). It was implemented in 4 cities of 2 counties from the north-west part of Romania (Cluj and Alba). In each county, the study was implemented in two cities—the city capital of the county (Cluj-Napoca for Cluj county, and Alba Iulia for Alba county, respectively) as well as other two cities (Câmpia Turzii and Câmpeni, respectively). School principals of 8 schools from the 4 cities were contacted, informed about the objectives of the study and its characteristics, and asked if they were willing to allow the implementation of the study in their school; out of these 8 school principals, 7 of them accepted to their schools to participate in the project and provided the number of the 5th to 8th grade classes that could participate. The parents of the children from these schools were contacted through letters, informing them about the study, and informed consent was obtained from them regarding their children’s participation.

The study involved several phases—an evaluation of nutrition and weight management-related opinions and behaviors of pupils (T1), followed by implementation of school based educational activities for promotion of healthy nutrition and active lifestyle. The present paper is based on data collected during T1.

At T1, the assessment was performed at school. Children from the participating classes were informed that the study involved anthropometric measurements (weight and height) and they were asked to fill in a confidential questionnaire. At the same time, at the beginning of the questionnaire it was clearly stated both verbally and in writing that their participation was voluntary and all the data would be confidential, with only the research team having access to the collected information. Children filled in the questionnaire, included it in an envelope provided by the research team, wrote their names on it, and a member of the research team collected the envelopes. By filling in the questionnaire, children gave their informed consent, while those who did not want to participate did not fill in the questionnaire. No refusal was recorded, but some children whose consent were given did not fill in the questionnaires because they were absent from school during the days of assessment. Anthropometric measurements were also performed by members of the research team. The final sample included 880 children (445 girls and 435 boys) aged 10–15 years old from grades 5th to 8th (middle school classes) who received informed consent for their participation from their parents, and they also voluntarily agreed to fill in the questionnaires. 

The parents of the participating students were also invited to fill in a confidential questionnaire sent through letters. As in the case of the children, they were explained about the confidentiality of their answers and that their participation in the program was voluntary. A total of 665 parents who filled in questionnaires returned them. 

### 2.2. Instruments for Data Collection

#### 2.2.1. Anthropometric Measurements

Children’s weight and height were measured in the classroom by a trained member of the research team in a standardized manner using a digital step scale and a portable stadiometer. The accuracy of the measurements was 0.1 kg and 0.5 cm, respectively. Children were measured barefoot, in light clothing. 

#### 2.2.2. Questionnaires

The study used a questionnaire which was developed based on data from the literature and on previous questionnaires developed and tested in several studies from Romania [27,51]. It included several sections that investigated nutrition, physical activity, weight management, and other health risk behaviors among participating children. The present study included the following aspects: Demographic characteristics such as Age and Gender.Perceived weight was assessed through the question *“How do you consider your weight?”* with three possible responses: Too big, Too low, or Normal.Desired weight: children’s weight management currently, with the following options: Lose weight, Gain weight, Stay at my current weight, or None of the above.Weight loss methods used in the last year by children who tried to lose weight: Food restrictions, Sport, Vomiting, Self-medication with diet pills, Slimming tea, or Massage (methods portrayed by different advertisements as ways to lose weight).Types of food restrictions used in the last year: Less food, Fewer calories, Skipping meals, Reduction of fat consumption, Less sugar, or Less carbohydrates.Physical activity (PA) performed in the last week was assessed through 2 items. The first item referred to the frequency and the possible options ranged from Never to 7 days a week. The second item assessed the duration of physical activity performed during those days. The average time dedicated to PA was calculated by multiplying the frequency by the duration (in minutes) and dividing by 7.Hours of sleep during the week and on weekends (responses ranged to 7 h or less to More than 10 h/day).Bullying—As a single item, children were asked if they have been bullied by others (bad words, laughter, shouting, or threats towards them) in the last 3 months because of their physical appearance or weight. Responses rated to a 4-point scale (Never, Once, 2–3 times, or More than 3 times).Cyber-bulling—Separate from the previous question, children were asked if they have been bullied through phone, e-mail, or social media by others (insults, bad words, or threats towards them) in the last 3 months because of their physical appearance or weight. Responses rated to a 4-point scale (Never, Once, 2–3 times, or More than 3 times).If children have ever been excluded from the group (do not talk, ignore, or avoid them) in the last 3 months because of their physical appearance or weight. Responses rated to a 4-point scale (Never, Once, 2–3 times, or More than 3 times).If children discussed their weight with their parents in the last year (Yes/No).

The parents’ questionnaire assessed their perception about their child’s weight by asking them “How do you find your child’s weight?” (An appropriate weight for his/her age, Too big, or Too small) and whether they have talked to a healthcare professional about their child’s weight in the last year (Yes/No).

### 2.3. Data Analyses

Actual weight of the children was assessed by calculating BMI (Body Mass Index) within the formula BMI = weight (kg)/height (m)^2^. Children’s BMI was categorized as Normal weight, Overweight, Obese, or Underweight, considering age and gender (Z-scores), according to World Health Organization (WHO) recommendations: overweight: >+1SD (equivalent to BMI 25 kg/m^2^ at 19 years); obesity: >+2SD (equivalent to BMI 30 kg/m^2^ at 19 years); and thinness: <−2SD [52]. Children’s weight status was afterwards collapsed into three categories: Underweight, Normal weight, and Excess weight (overweight and obese children). 

The prevalence of the investigated issues was calculated for the whole sample, as well as separately for boys and girls according to BMI classification.

Logistic regression analyses were used to examine the relationship between several factors and higher BMI in children (coded as 0 = Normal weight, 1 = Excess weight), as well as the differences in misperceptions of children’s weight (for normal weight children we investigated two subcategories: correct estimation vs. incorrect underestimation and correct estimation vs. incorrect overestimation, while for excess weight children, we evaluated correct estimation vs. incorrect underestimation); correct estimation was considered when the perceived weight by children was in the same category as the actual weight, while when the perceived weight was in a higher or lower category than that of the actual weight, it was considered an overestimation or underestimation, respectively. The independent variables considered were gender (girls = 1, boys = 0), age, personal estimation of weight (only for BMI), and parental estimation regarding weight based on whether or not their perception of their children’s weight status matched with the children’s actual BMI (Correct = 1, Incorrect = 0), children’s weight management (Attempts to lose weight coded as Yes = 1, No = 0 and Attempts to gain weight coded as Yes = 1, No = 0), discussion between children and their parents on weight issues (Yes = 1, No = 0), discussion between parents and a healthcare professional regarding their child’s weight (Yes = 1, No = 0), time spent on physical activities (<1 h = 1, 1–2 h = 2, >2 h = 3), and the presence of bulling, cyber bulling, or being excluded from the group in the last three months (Never = 0, <3 times = 1, ≥3 times = 2). We estimated adjusted odds ratio (OR) and 95% confidence interval (95% CI). The significance level was set to *p* < 0.05. Underweight participants were excluded from logistic regressions, due to their low number (N = 65). 

All statistical analyses were performed using IBM SPSS Statistics 26 (IBM Corporation, Armonk, NY, USA)

## 3. Results

### 3.1. Real Weight, Perceived Weight, and Desired Weight-Opinions of Children and Parents

Table 1 shows the opinions of children and their parents regarding real weight, perceived weight, and desired weight. The actual weight of children recruited into the study, according to the BMI and cutoffs stipulated by the WHO, showed that 61.0% of them (N = 537) were of normal weight, 7.4% (N = 65) were underweight, and 31.6% (N = 278) were of excess weight (overweight or obese).

Children’s assessments regarding their perceived weight showed that more than half of normal weight and excess weight children, as well as 40% of underweight children estimated their weight correctly. 

Moreover, about 30% of normal weight children and more than half of overweight children reported attempts to lose weight in the previous year, whereas about 45% of underweight children and about 15% of normal weight children tried to gain weight. 

Among weight loss methods, restrictive diets and physical activities were the most practiced by children, especially for overweight students (41.7% and 49.6%, respectively), whereas fasting, eating fewer calories, eating food low in fat, and eating less sugar and less carbohydrates were the dietary restrictions most frequently reported. 

On the other hand, there were some differences in measured BMI regarding average time spent on physical activities. Excess weight children were the most inactive of all, with 81.6% of them investing less than 60 min/day in PA. In addition, only 25% of children from the other 2 subgroups, followed the WHO recommendations of 60 min per day physical activity [53].

Regarding time spent during night sleep, there were no differences in measured BMI. About 70% of students slept between 8 and 10 h during the week. About 60% of them slept just as much during the weekend and more than 15% slept more than 10 h.

The prevalence of bullying among children shows that 39% of overweight children, followed by 33% of underweight children, have been verbally aggressed because of their physical appearance at least once in the last 3 months. Additionally, overweight children were the most frequently aggressed on social media (22.3%) and excluded from groups (23.5%). 

Furthermore, 22.9% of normal weight children, 16.9% of underweight children, and 32.4% of overweight children declared they have discussed their weight with their families in the previous year. 

As far as parents are concerned, a third correctly estimated the weight of their underweight children and overweight children, and a majority of them correctly estimated the weight of their normal weight children. A total of 60% of parents of overweight children and 45% of parents of underweight children misperceived their child’s weight as normal, whereas 21% of parents of underweight children and 7.4% of parents of normal weight children estimated their weight as too high. 

Almost half of parents of normal weight children and more than half of parents of underweight and overweight children declared they spoke with a healthcare professional about their children’s weight in the previous year.

Table 2 shows the results separately, by gender. The data presented in these tables show that boys estimated their real weight better among normal weight children (71.2% vs. 63.3%) and underweight children (46.0% vs. 32.1%), while girls estimated their weight better among overweight children (65.1% vs. 51.2%).

Regarding body weight management, girls declared more attempts to lose weight compared to boys, especially among overweight children (74.5% vs. 59.3%), followed by normal weight children (34.4% vs. 22.6%). However, underweight boys declared more attempts to gain weight, by twice as much as underweight girls. Regarding weight loss methods, girls outnumbered boys, but both used restrictive diets and physical activities as weight control behaviors. Nevertheless, in all three BMI subcategories, boys mainly resorted to sports. Girls invested less time in PA than boys, with underweight females and excess weight males being the most inactive. There were no differences in measured BMI and gender regarding time spent during night sleep. On the other hand, girls spoke more frequently with their family about body weight.

### 3.2. Characteristics of Children in Different Weight Categories

The results of the bivariate logistic regression showed that participants with excess weight versus normal weight were less likely to be females, were less likely to correctly estimate their weight, expressed attempts to lose weight in the previous year, tended to discuss more with their parents about their weight, and were more likely exposed to bullying and exclusion from groups. Moreover, their parents were less likely to appreciate with accuracy their children’s weight (see Table 3).

### 3.3. Factors Which Influence the Correct Estimation of Children’s Own Real Weight

We first analyzed children with normal weight. The most common bias regarding body image self-perception was that more than 20% of participants believed they were heavier than they were, followed by 11.7% students who underestimated their weight. However, less than 20% of parents misperceived their child’s weight (See Table 1).

Table 3 shows the factors associated with children’ self-perceived body weight. Compared to children who underestimated their size, those whose weight was in accordance with their actual BMI expressed needs for losing weight in the previous year and were less likely to be exposed to bullying, cyberbullying, and exclusion from groups. Moreover, their parents were more likely to correctly perceive their children’s weight. On the other hand, children who perceived their weight accurately, in comparison to children who overestimated their size, were less likely to be females, declared attempts to gain weight in the previous year and were less exposed to bullying, cyberbullying, and exclusions from groups. In addition to these, they were less likely to discuss their weight with their parents. Additionally, their parents tended to correctly perceive their children’s size.

Secondly, we evaluated children with excess weight. Among them, about 44% underestimated their weight, while more than 60% of their parents misperceived their child’s size (See Table 1).

Among children with excess weight, overweight children who accurately estimated their weight, in contrast to overweight children who underestimated their weight, were more likely to be girls, tried to lose weight in the previous year, talked with their parents about their weight, and were more likely to be bullied. Moreover, their parents correctly assessed their children’s weight (See Table 3).

## 4. Discussion

The aim of this study was to examine the child-parent dyad related to the children’s perceived and desired weight in contrast to the children’s actual weight. Additionally, within this study, we explored the relationships between under- and over-estimation in normal and heavy weight children, looking to both the weight-related experiences and behaviors of themselves and their parents. To our knowledge, this study would be the second research conducted in Romania that evaluated these issues [27]. Compared to the previous study which was carried out in 2013 on a group of 344 students from a city of Romania, our study was conducted in 2019 and included a larger sample of children from several cities of Romania which allowed separate analyses for girls and boys. Moreover, it included additional topics such as physical activity, hours of sleep, and types of bullying that have not been assessed before, and we examined their relationship to the actual weight and perceived weight.

Our study showed that 31.6% of the participating children were overweight or obese. Another study performed in 2016 on a large representative sample from Northwestern Romania among 7–18 year old children showed a prevalence of overweight and obesity of about 30% [50]. In addition, a small percentage of children recruited in our study were underweight (7.4%).

Moreover, it was observed that there was a discordance between measured BMI and perceived weight among students and their parents. Almost half of the children with excess weight underestimated their size, followed by normal weight children in 11.7% of cases. Similar results were found in the previous study conducted in 2013 in Romania [27]. In addition, 60% of underweight pupils overestimated their weight, followed by one-fifth of normal weight children who estimated their size heavier than it was. This bias was more frequently met in girls.

Discrepancies were noticed among children’s weight change initiatives, too. Healthy weight children reported attempts to lose weight in about 30% of cases. As it was expected, girls attempted weight loss more frequently than boys. These findings might have two explanations. Firstly, they misjudged their size by overestimating their weight and secondly, unnecessary weight control behaviors might be connected to body-image dissatisfaction. Children tend to internalize the concept of the ideal body weight, as they are susceptible to such messages from peers and social media, and to compare their own weight against these standards. Our results are similar with other previous studies. For example in his study, Fredrikson et al. outlined that females are influenced by a thin ideal body and differences between an idealized thin body and their actual BMI provoke dissatisfaction and a desire to lose weight [14]. A systematic review conducted by Duarte and her colleagues concluded that body weight dissatisfaction is frequently reported in adolescent population, regardless of the accuracy of their weight [54].

On the other hand, unlike the previous Romanian study, 20% of underweight children reported attempts to lose weight in the previous year as well [27]. This finding could also be attributed to body weight dissatisfaction. Body image distortion includes negative beliefs concerning body shape and appearance (cognitions), misperception (incorrect evaluation of individual own’s size, shape, and weight), an affective compound, and a behavioral impairment (such as body checking, unhealthy weight-related behaviors, dieting, and isolation) which are core factors for the development of mental health problems, including disorder eating or even body dysmorphic disorder [55].

Moreover, our study revealed that 15% of normal weight participants and 5.4% of overweight participants tried to gain weight the year before. A potential explanation would be that children copy models seen from peers or family members, without considering their physiological needs.

Nevertheless, almost half of underweight children tried to gain weight in the previous year, whereas 65.1% of overweight children self-reported attempts to lose weight the year before, which might suggest that the desired weight is associated to a healthy weight. Based on gender, underweight boys were more concerned to gain weight and overweight girls to lose weight. A likely explanation would be that underweight males tried to gain muscle mass, whereas excess weight females tried to achieve a slim body.

The most common weight loss methods reported were sports and diet restrictions as in the previous study conducted in Romania, but in addition to this, it could be observed that these were found especially in overweight children and girls. Unlike females, males prefer sports more. However, around one-fifth of children do not follow the WHO recommendations, which is at least 60 min per day of physical activity [53]. In fact, Maoyong Fan in his study showed that, despite the fact that correct self-perception of being overweight was strongly associated with weight-loss intentions, children failed to engage in healthy diet habits and moderate or vigorous exercises [56]. In addition, Duarte’s review showed a relationship between unhealthy eating behaviors (low fruits and vegetables intake, consumption of soft drinks, and low breakfast consumption) and lack of physical activity with body image dissatisfaction [54].

Additionally, results of our research were in line with other previous studies that documented overweight children were victims of bullying, aggressions on social media, and peer exclusions [57,58]. Despite the valuable contributions from the obesity perspective, there is little research on bullying from an underweight individual’s view [59]. Evidence showed that preference for “normal weight” leads to prejudice, exclusion, and aggression not only towards obese children, but underweight children, too [60]. Our study showed that heavy weight children, followed by their underweight counterparts have been victims of bullying at least once (39.2% vs. 33.8%, respectively). Moreover, cyberbullying and exclusion from groups were more frequently declared by overweight children. Similar results were found in Wilson’s study in which he examined the relationship between actual BMI, perceived body weight, and bullying [61]. Possible reasons could be that girls are deemed to be slim in order to be socially accepted, whereas underweight boys are considered weak [62].

In addition to these aspects, our study investigated parents’ perception regarding their child’s body weight. Similar to other studies, our results show that a large proportion of parents are unaware of their children’s actual weight. A total of 62.9% of parents of children with excess weight underestimate the weight of their child, with 20% more than the percentage observed by the previous Romanian study [27]. Moreover, they misperceive the size of their child more frequently than overweight children themselves (43.5%).

Parents’ bias regarding children’s body weight could be explained by several factors. First, since the prevalence of obesity in children increased lately, the view of the community regarding excess weight has been desensitized and nowadays it is perceived as normal [63,64]. Due to lack of knowledge, parents tend to estimate their children’s body weight through visual comparison with other children, who subjectively are perceived as overweight. Second, parents might be reluctant to accept that their child as overweight or obese, because of social pressures [64]. On the other hand, parents’ misperception is rather an emotional evaluation, rather than a cognitive bias [65].

The literature suggests that parents’ role in prevention of their children’s excess weight is influenced by the fact that they fail to recognize accurately their child’s body weight [60]. Healthcare professionals should help parents in identifying and correcting their misperceptions regarding their children’s size, as well as draw their attention to the future health risks that derive from obesity [12,60]. In our case, one of two parents whose children were with excess weight spoke with a healthcare professional about their children’s weight status in the previous year. Additionally, communication between children and their parents on weight issues were more frequently reported among excess weight children (especially among children with excess weight who perceived their weight correctly) and among normal weight children who overestimated their weight. There were noticed gender differences, with girls declaring they have spoken more often with parents about weight, compared to boys. As other studies showed, one possible explanation could be that females receive socio-cultural messages about weight loss from their family, whereas boys perceive pressures in increasing muscle mass [66,67]. Second, as girls learn about these socio-cultural standards early in life, they might be aware and influenced by them during middle school and maybe even earlier [68,69,70]. Moreover, girls might be more interested and open to discussing weight issues with their parents, because of their uncertainties, seeking information or a desire to share their concerns; in our study, normal weight girls were more likely to think their weight was too high compared to normal weight boys, while those who overestimated their size reported more often talking with their parents on this issue. On the other hand, data from other countries suggest that boys show an interest in increasing the size of their upper body late in puberty [71].

This study is subject to several limitations. First, our sample included children from four urban settings from the north-west part of Romania; therefore, the generalization of the provided information to the whole country is limited. Second, we did not assess the relationship between socio-economic status and children’s weight. Third, data on bullying and physical activity were based on a self-reported questionnaire that could influence our findings due to memory bias. Perceptions of children’s weight were assessed through questioning teenagers how they estimate their weight (too big, too small, or normal), instead of using specific tools for this issue. Moreover, due to the cross-sectional design of the study, we could not determine the longitudinal consequences of weight behaviors on children’s BMI nor explore the long-term parental effects on children’s physical and mental wellbeing. Moving further, the preliminary results on bullying give impetus to further research on the effects of weight stigma on children’s mental health and its relationship to disordered eating. In the end, the sample of parents was modest, and our data was based on their statements. Further studies should focus more on parents’ motivation to participate and assess differences in gender perceptions. Moreover, a closer look at communication between children and parents on these issues could provide useful information for health education programs.

The results have implications for future health promotion activities and research. Concentrated efforts should be directed towards weight related issues in which both children and parents should be involved actively. These initiatives need to be included into a multidimensional approach with children, parents, schoolteachers, and healthcare professionals, as well as community working closely together. Implementation of educational programs in schools as well as designing appropriate education, information actions, and facilitating access to educational resources could help pupils understand the principles for a healthy diet, dispel the myth of an ideal body that children want to achieve, and prevent and reduce the trend of childhood obesity. On the other side, the results emphasize the importance of including these issues in children’s health education curricula, and the prevention and reduction of bullying being relevant in relation to the physical and emotional wellbeing of children, regardless of their weight.

Moreover, health care professionals should aim to raise parental awareness through counseling. For example, the healthcare professional could use the BMI as a screening tool to monitor the child’s weight status during periodical consultations, and explain and reflect with parents the risks of obesity and the appropriate tools for communicating with their children, in order to avoid inappropriate teasing and stigma or pressure to lose weight, which could have a negative effect on both nutrition and children’ weight related behavior, as well as their mental health and wellbeing, as other studies also suggest [51,55]. Moreover, physicians should support parents to encourage a healthy lifestyle inside the family.

## 5. Conclusions

Our study identified misperceptions among children and their parents regarding children’s actual weight, which may negatively influence their current lifestyle with risk of delayed intervention on body weight management and development of future health complications derived from obesity and weight dissatisfaction. Factors such as body mass index, gender, weight related behaviors, parents’ estimation about their children’ weight, discussions inside family on weight topics, and bullying, cyberbullying and exclusion from groups were associated with misperceptions.

The study provides useful information for health education activities for both children and their parents on the proper management of children’s body weight, as well as efforts to reduce weight related stigma and bullying. Future research should pay attention to the development and evaluation of educational programs and measures to promote adequate weight, a healthy lifestyle, and a good quality of life among Romanian children.

## Figures and Tables

**Table 1 ijerph-19-03502-t001:** Prevalence of body weight perception and weight-related behaviors according to BMI among the whole sample.

	Actual Weight (BMI)		
	Underweight N = 65 7.4%	Normal Weight N = 537 61.0%	Excess Weight N = 278 31.6%
Perceived weight by children			
Under weight	40%	11.7%	6.8%
Normal weight	40%	66.7%	36.7%
Excess weight	20%	21.6%	56.5%
Children’s behavior regarding their weight			
Attempts to lose weight	20%	29.4%	65.1%
Attempts to gain weight	44.6%	14.9%	5.4%
None	35.4%	55.7%	29.5%
Weight loss methods (multiple response)			
Purging	0.0%	2.0%	2.9%
Diet pills	0.0%	1.5%	4.3%
Food restrictions	15.4%	25.9%	41.7%
Slimming tea	0.0%	2.8%	2.9%
Sport	23.1%	36.1%	49.6%
Massage	1.5%	0.9%	1.4%
None	72.3%	52.5%	33.1%
Types of food restrictions (multiple response) *	N = 10	N = 139	N = 116
Less food	70.0%	48.2%	58.6%
Fewer calories	30.0%	53.2%	55.2%
Skipping meals	30.0%	22.3%	25.9%
Fat reduction	30.0%	52.5%	53.4%
Less sugar	50.0%	64.7%	59.5%
Less carbohydrates (bread, pasta, potato)	40.0%	52.5%	57.8%
Physical time activity/day **	N = 62	N = 524	N = 272
<1 h	77.4%	75.6%	81.6%
1–2 h	14.5%	17.7%	13.2%
>2 h	8.1%	6.7%	5.2%
Hours of sleep/night during the week			
≤7 h	20.0%	27.2%	27.3%
8–10 h	75.4%	67.4%	69.1%
>10 h	4.6%	5.4%	3.6%
Hours of sleep/night during the weekend			
≤7 h	20.0%	16.4%	20.9%
8–10 h	63.0%	62.2%	64.0%
>10 h	17.0%	21.4%	15.1%
Bullying			
Never	66.2%	74.5%	60.8%
Up to 3 times	30.8%	19.9%	29.5%
More than 3 times	3.0%	5.6%	9.7%
Cyber bullying			
Never	87.7%	82.5%	77.7%
Up to 3 times	10.8%	14.3%	17.3%
More than 3 times	1.5%	3.2%	5.0%
Exclusion from the group			
Never	87.7%	83.2%	76.5%
Up to 3 times	10.8%	14.2%	19.5%
More than 3 times	1.5%	2.6%	4.0%
Discussion on weight between children and their parents			
Yes	16.9%	22.9%	32.4%
No	83.1%	77.1%	67.6%
Perceived weight of children by parents			
Under weight	34.9%	6.8%	2.9%
Normal weight	44.2%	85.8%	60.0%
Excess weight	20.9%	7.4%	37.1%
Discussions between parents and a healthcare professional regarding children’s weight			
Yes	58.7%	48.4%	50.5%
No	41.3%	51.6%	49.5%

* The values were calculated according to the number of children who stated they used food restrictions. ** The values were calculated according to the number of children who stated they performed physical activity.

**Table 2 ijerph-19-03502-t002:** Prevalence of body weight perception and weight-related behaviors on gender and BMI.

	BMI Category					
	Underweight		Normal Weight		Excess Weight	
	Girls N = 28	Boys N = 37	Girls N = 311	Boys N = 226	Girls N = 106	Boys N = 172
Perceived weight by children						
Under weight	32.1%	46.0%	10.0%	14.2%	3.8%	8.7%
Normal weight	42.9%	37.8%	63.3%	71.2%	31.1%	40.1%
Excess weight	25.0%	16.2%	26.7%	14.6%	65.1%	51.2%
Children’s behavior regarding their weight						
Attempts to lose weight	21.4%	18.9%	34.4%	22.6%	74.5%	59.3%
Attempts to gain weight	28.6%	56.8%	13.2%	17.3%	2.0%	7.6%
None	50.0%	24.3%	52.4%	42.8%	23.5%	33.1%
Weight loss methods (multiple response)						
Purging	0.0%	0.0%	1.0%	3.5%	1.9%	3.5%
Diet pills	0.0%	0.0%	1.3%	1.8%	4.7%	4.1%
Food restrictions	17.9%	13.4%	30.5%	19.5%	60.4%	30.2%
Slimming tea	0.0%	0.0%	1.9%	4.0%	2.8%	2.9%
Sport	28.6%	18.9%	40.8%	29.6%	58.5%	44.2%
Massage	0.0%	2.7%	0.6%	1.3%	1.9%	1.2%
None	71.4%	73.0%	49.2%	57.1%	25.5%	37.8%
Types of food restrictions (multiple response) *	N = 5	N = 5	N = 95	N = 44	N = 64	N = 52
Less food	60.0%	80.0%	51.6%	40.9%	57.8%	59.6%
Fewer calories	20.0%	40.0%	51.6%	56.8%	51.6%	59.6%
Skipping meals	20.0%	40.0%	22.1%	22.7%	28.1%	23.1%
Fat reduction	20.0%	40.0%	58.9%	38.6%	56.3%	50.0%
Less sugar	60.0%	40.0%	67.4%	59.1%	62.5%	55.8%
Less carbohydrates (bread, pasta, potato)	40.0%	40.0%	55.8%	45.5%	67.2%	46.2%
Physical time activity/day **	N = 27	N = 35	N = 306	N = 218	N = 104	N = 168
<1 h	92.6%	65.7%	78.8%	71.1%	89.4%	76.8%
1–2 h	7.4%	20.0%	16.7%	19.3%	8.7%	16.1%
>2 h	0.0%	14.3%	4.5%	9.6%	1.9%	7.1%
Hours of sleep/night during the week						
≤7 h	21.4%	19.0%	28.3%	25.7%	31.1%	25.0%
8–10 h	71.5%	78.3%	66.9%	68.1%	65.1%	71.5%
>10 h	7.1%	2.7%	4.8%	6.2%	3.8%	3.5%
Hours of sleep/night during the weekend						
≤7 h	17.9%	21.6%	14.8%	18.6%	14.2%	25.0%
8–10 h	67.8%	59.5%	64.0%	59.7%	68.8%	61%
>10 h	14.3%	18.9%	21.2%	21.7%	17.0%	14%
Bullying						
Never	75.0%	59.5%	74.0%	75.2%	57.5%	62.8%
Up to 3 times	21.4%	37.8%	18.4%	22.1%	32.1%	27.9%
More than 3 times	3.6%	2.7%	7.6%	2.7%	10.4%	9.3%
Cyber bullying						
Never	85.7%	89.2%	83.0%	81.9%	80.2%	76.2%
Up to 3 times	10.7%	10.8%	13.8%	15.0%	16.0%	18.0%
More than 3 times	3.6%	0.0%	3.2%	3.1%	3.8%	5.8%
Exclusion from the group						
Never	82.2%	91.1%	85.5%	80.1%	78.3%	75.6%
Up to 3 times	14.2%	8.1%	11.0%	18.6%	17.0%	20.9%
More than 3 times	3.6%	0.0%	3.5%	1.3%	4.7%	3.5%
Discussion on weight between children and their parents						
Yes	25.0%	10.8%	25.7%	19.0%	39.6%	27.9%
No	75.0%	89.2%	74.3%	81.0%	60.4%	72.1%
Perceived weight of children by parents						
Under weight	28.6%	40.9%	5.8%	8.6%	1.2%	4.0%
Normal weight	52.4%	36.4%	86.4%	84.7%	59.3%	60.5%
Excess weight	19.0%	22.7%	7.8%	6.7%	39.5%	35.5%
Discussions between parents and a healthcare professional regarding children’s weight						
Yes	57.1%	60.0%	45.8%	52.6%	44.6%	54.1%
No	42.9%	40.0%	54.2%	47.4%	55.4%	45.9%

* The values were calculated according to the number of children who stated they used food restrictions. ** The values were calculated according to the number of children who stated they performed physical activity.

**Table 3 ijerph-19-03502-t003:** Factors associated with BMI and children’s self-perceived body weight (correct-incorrect estimation).

	BMI Category	Normal Weight Children		Excess Weight Children
	Excess Weight vs. Normal Weight	Underestimation	Overestimation	Underestimation
	OR (95%CI)	OR (95%CI)	OR (95%CI)	OR (95%CI)
Age	0.88 (0.77–1.00)	0.98 (0.77–1.24)	1.04 (0.87–1.25)	1.01 (0.82–1.24)
Gender				
Female (vs. Male)	0.44 (0.33–0.60) ***	1.26 (0.73–2.15)	0.48 (0.30–0.76) **	1.78 (1.08–2.93) *
Weight estimation by children				
Correct (vs. Incorrect)	0.64 (0.48–0.87) ***	–	–	–
Attempts to lose weight in the last year				
Yes (vs. No)	4.47 (3.28–6.09) ***	11.4 (1.56–84.6) *	0.02 (0.01–0.05) ***	22.3 (11.4–43.5) ***
Attempts to gain weight in the last year				
Yes (vs. No)	0.32 (0.18–0.57) ***	0.02 (0.01–0.05) ***	5.21 (1.22–22.1) *	0.04 (0.00–0.37) **
Discussion between children and parents regarding weight				
Yes (vs. No)	1.61 (1.16–2.22) ***	0.89 (0.44–1.82)	0.21 (0.13–0.33) ***	2.93 (1.69–5.07) ***
Weight estimation by parents				
Correct (vs. Incorrect)	0.09 (0.06–0.14) ***	4.35 (1.96–9.67) ***	6.35 (3.25–12.3) ***	5.77 (2.93–11.3) ***
Discussions between parents and a healthcare professional regarding children’s weight				
Yes (vs. No)	1.08 (0.78–1.51)	0.70 (0.37–1.30)	1.03 (0.62–1.72)	0.99 (0.58–1.70)
Physical activity	0.78 (0.59–1.02)	1.05 (0.65–1.67)	1.05 (0.73–1.52)	0.90 (0.57–1.40)
Bullying	1.58 (1.25–1.99) ***	0.35 (0.23–0.54) ***	0.45 (0.31–0.64) ***	1.75 (1.20–2.57) **
Cyber bullying	1.29 (0.97–1.70)	0.29 (0.18–0.46) ***	0.53 (0.34–0.83) **	1.43 (0.90–2.26)
Exclusion from the group	1.38 (1.03–1.85) *	0.29 (0.17–0.50) ***	0.50 (0.32–0.78) **	1.11 (0.70–1.75)

* *p* < 0.05; ** *p* < 0.01; *** *p* < 0.001.

## Data Availability

The data presented in this study are available on justified cases from the first author.

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
