# Peer review of "Assessment of Actual Weight, Perceived Weight and Desired Weight of Romanian School Children-Opinions and Practices of Children and Their Parents"

_ijerph, 2022, doi:10.3390/ijerph19063502_

Round 1

Reviewer 1 Report

The present study describes a cross-sectional study conducted among Romanian school children, in which parent and child dyads rated the children’s perceived and desired weight, and researchers also measured the children’s actual weight. Relationships between weight under- and over-estimation and weight-related experiences and behaviours were explored. 

The study topic is interesting, and certainly it is nice to see research from outside North American samples. However, the framing of this study is very problematic. The researchers are approaching this issue from the premise that weight misperception is a problem as it likely interferes with dieting attempts and weight loss. First, effects on dieting attempts are only pertinent in higher-weight children who think they are normal weight. Effects on ‘underweight’ children who think they are ‘normal’ or ‘overweight’ are likely to be less straightforward as elements of body dysmorphia and eating disorder cognitions and behaviours are likely to complicate simple discussions around weight-related behaviours. These two groups should not be treated equally. 

However, more important, while it might seem intuitive that higher-weight children should be told they are too heavy and be encouraged to diet, the literature on weight misperception quite clearly indicates that being labelled as ‘overweight,’ ‘fat,’ or ‘too heavy’ is linked with worse long-term health outcomes, not better. This includes greater risk of disordered eating and, paradoxically, weight gain. These results hold even controlling for baseline weight or accuracy of the label to the child’s actual body weight. Likewise, accurate misperception of ‘overweight’ versus misperception by ‘overweight’ youngsters who believe they are ‘not overweight’ is associated with worse outcomes. For some refs in this area, see for example, Hunger & Tomiyama, 2014 (doi:10.1001/jamapediatrics.2014.122) and 2018 (https://doi.org/10.1016/j.jadohealth.2017.12.016), Robinson et al., 2020 (https://doi.org/10.1038/s41366-020-0587-6), and Kline, 2015 (http://dx.doi.org/10.1108/YC-12-2014-00495). For what it’s worth, such findings also reproduce for adults, and not only in Western samples (e.g., Kim et al., 2018, https://doi.org/10.1371/journal.pone.0198841). Labelling is also associated with fewer health behaviours and great levels of unhealthy behaviours. In contrast, weight satisfaction (vs dissatisfaction), even in higher-weight individuals, is linked to better mental and physical health and better long-term health outcomes, including reduced incident of diabetes and other chronic conditions (see e.g, Muennig et al., 2008, doi:10.2105/AJPH.2007.114769; Blake et al., 2013, doi: 10.1155/2013/291371; Wirth et al., 2015, http://dx.doi.org/10.1016/j.bodyim.2014.11.003; and 2014, doi:10.1037/hea0000058). For a review of some of this literature, including the reasons why the obesity  dieting  improved health paradigm is not supported by the evidence, see a recent review by Hunger et al., 2020 (DOI: 10.1111/sipr.12062). This review is actually not exhaustive on the literature around labelling/weight perception, and is not focussed on children’s health (although it discusses BMI surveillance, weight report cards, labelling etc). I do not have time to search for other papers on weight misperception, but the authors should consider themselves under a moral obligation to do so if they intend to publish this work. Your one mention of this conundrum is in the Discussion – your ref 40 – says the same, correct ‘overweight’ perception is associated with greater weight loss intentions but not with healthy behaviours. 

In your discussion, you note as a possible explanation for parental misperceptions: “Last but not least, parents’ concern is that their child is in good health, meaning being happy and active, rather than relating to the body size per se [45,48].” This is absolutely how it should be. Similar issues to those associated with labelling/correct perception arise with parental input, comments and food-control behaviours around children’s weight. These also tend to be associated with more disordered eating, negative psychological outcomes, and weight gain – again, the authors are encouraged to explore this literature before publishing this work – the suggestion that parental intercession in child weight-control is a positive goal should be reconsidered. There is no doubt that misperception can mean fewer dieting attempts and less parental effort to get children to lose weight – that this is problematic is only assumed, based on equating weight with health. You say that “recognition is a critical step in obesity prevention.” We do not have data that suggest telling children they are overweight, or getting their parents to notice they are overweight, and do something about this provides any form of long-term positive outcome.

Even more extensive is the well–documented negative impact of weight stigma on child and adolescent health, including, again, weight gain, but also anxiety, depression, self-harm, and suicidality independent of starting weight/BMI. Problematising fat bodies inevitably leads to this negative perception and stigmatisation of fat children (and adults).

Assuming the authors are willing to reframe their study, I also have a few minor comments on the study itself and reporting of the results that I hope the authors find helpful. These are intended to help improve clarity and do not reflect on the quality of the paper: 

•    School is not a ‘demographic characteristic’ as such. Was this used in some way as a proxy for socioeconomic status? Was SES measured and controlled for? It might be helpful to include a summary of the sample demographics.
•    Desired weight: what does ‘none of the above’ refer to? Was this a ‘prefer not to answer’ option?
•    Slimming tea and massage are not common forms of weight control behaviours in most countries that I am aware of. This might be a local phenomenon? It might be worth expanding on this a little – in particular, I can imagine that ‘slimming tea’ may be supposed to have some fat burning compounds, but how is massage supposed to lead to weight loss?
•    The descriptions of physical activity are not consistent. Defined as ‘at least 10 minutes’ but ‘available responses ranged from half an hour to more than 2 hours.’ Do you mean ‘actual responses’ rather than ‘available responses’? Please clarify.
•    The bullying question asks about bullying ‘because of their physical appearance or because of their weight.’ These are not necessarily the same thing. Were these items asked separately or was that a single item? Additionally, was this question about in-person bullying only, with the following person asking about cyberbullying, or was the first question about ALL bullying and the second question just one component of this? Please clarify.
•    Children were asked if they had discussed their weight with their parents in the previous year. How accurate is this likely to be? Especially in younger children. In addition to the issue of recall, the time frame is also likely to pose problems to most children. Also, responses are likely systematically biased, with children who have ‘problem’ weight, more likely to remember such a family discussion. 
•    In data analysis, what is meant by ‘underweight and thin’? Is this one category? Is this a translation issue?
•    Frequency of cyberbullying: either the ‘less than’ or the ‘more than’ option should include ‘equal to’ three times.
•    Bottom of page 4: You use the sentence, “Parents’ assessment regarding children’s body size was similar. Their possible responses were dichotomized into two categories (Correct/Incorrect estimation), whether their perception on child’s weight status matched or not with children’s actual BMI.” This is a beautifully succinct description that could be use for both children and parents. The preceding three paragraphs saying essentially the same thing are quite repetitive and confusing.
•    Logistic regression coding. 0 = NW, 1 = EW. What about underweight? Were they excluded because of low numbers? Please clarify. 
•    In the table note for T1, you indicate that the asterisk marks that type of restricted eating analysis was reported for those who indicated food restriction. Please insert the N either in the table note or in the table itself.
•    Tables 2 and 3. I think it would be more helpful to see these side by side, rather than  having to go backwards and forwards between them. Could you please combine into a single table.
•    Just a query – purging (vomiting) in boys was almost double that in girls? Is that correct?
•    End of section 3.1, girls more concerned to discuss weight with family: I think it’s more likely to be the other way round – parents are more concerned about discussing weight with female children than male children – this is consistent with body ideal norms being enforced more strongly in girls, greater body objectification and self-objectification in girls, and girls experiencing more weight stigma and at lower BMI than boys. Maybe mention this in the discussion – this gender difference indicates that parental weight talk is definitely more about social norms than about health, otherwise it’d be the same for both.
•    Section 3.2 heading is misleading. Suggest changing it to, “Characteristics of children in different weight categories”
•    Table 4 is really confusing to follow. There must be a better way to report these findings? 
•    Smaller adjustments to Table 4 that might help:

    • Remove the single bullet point for non-significant findings. The absence of the traditional asterisks is indicative of lack of significance.
    • BMI should be ‘BMI category’
    • Also, table headings are confusing by inferring a reference category that is not necessary. Perhaps just use Normal weight children ‘underestimators’ and ‘overestimaters,’ and Excess weight children ‘underestimaters.’
    • Similarly, instead of using two rows and including a reference row ‘1’, you could just put it in the variable name (e.g., “Female (vs. male)).
    • I had several other comments/queries but they might be due to confusion because of the double reference category (columns and rows) – fix that first and hopefully it’ll become clearer.
    • Are the WL attempt figures in the correct columns? Maybe just simplify headings and then double check all the numbers.
  • Discussion, “girls tended to lose weight more frequently than boys” – be careful here – correct to “girls attempted weight loss more frequently than boys.” The most common outcome of weight loss attempts is actually weight gain.
    •    % trying to gain weight: Weight gain attempts were more frequent in boys – is it possible this was about gaining muscle rather than fat? I can’t imagine any reason why a child would want to gain fat tissue because others in their family or among their peers were fatter. This needs to be better explained.
    •    The manuscript would also benefit from a proofread by a native English speaker.

Author Response

 Response letter

We thank you for your quick response and we would like to thank the reviewers for the through
feedback. We have adjusted our article according to their suggestions, and their work was invaluable
in revising our work.
We have added our answers to every question received from the reviewers in this letter below. We
have also highlighted the changes in our article as well as in this letter, under every question placed
by the reviewers.
Reviewer 1
1. The present study describes a cross-sectional study conducted among Romanian school
children, in which
parent and child dyads rated the children’s perceived and desired weight, and
researchers also measured the children’s actual weight. Relationships between weight under- and
over-estimation and weight-related experiences and behaviours were explored.
The study topic is interesting, and certainly it is nice to see research from outside North American
samples. However, the framing of this study is very problematic. The researchers are approaching
this issue from the premise that weight misperception is a problem as it likely interferes with dieting
attempts and weight loss. First, effects on dieting attempts are only pertinent in higher-weight children
who think they are normal weight.
Effects on ‘underweight’ children who think they are ‘normal’ or
‘overweight’ are likely to be less straightforward as elements of body dysmorphia and eating disorder
cognitions and behaviours are likely to complicate simple discussions around weight-related
behaviours. These two groups should not be treated equally.
Response:
Now, in the Discussion section, we treated separately the group of underweight children. Please see
below.
“On the other hand, unlike the previous study, 20% of underweight children reported attempts to lose
weight in the previous year as well. This finding could also be attributed to body weight
dissatisfaction. Body image distortion includes negative beliefs concerning body shape and
appearance (cognitions), misperception (incorrect evaluation of individual own’s size, shape, and
weight), an affective compound and a behavioural impairment (such as body checking, unhealthy
weight-related behaviours, dieting, isolation) which are core factors for the development of mental
health problems, including disorder eating or even body dysmorphic disorder [55].”
2. However, more important, while it might seem intuitive that higher-weight children should
be told they are too heavy and be encouraged to diet, the literature on weight misperception quite
clearly indicates that being labelled as ‘overweight,’ ‘fat,’ or ‘too heavy’ is linked with worse longterm health outcomes, not better. This includes greater risk of disordered eating and, paradoxically,
weight gain. These results hold even controlling for baseline weight or accuracy of the label to the
child’s actual body weight. Likewise, accurate misperception of ‘overweight’ versus misperception
by ‘overweight’ youngsters who believe they are ‘not overweight’ is associated with worse outcomes.
For some refs in this area, see for example, Hunger & Tomiyama, 2014

(doi:10.1001/jamapediatrics.2014.122) and 2018 (https://doi.org/10.1016/j.jadohealth.2017.12.016),
Robinson et al., 2020 (https://doi.org/10.1038/s41366-020-0587-6), and Kline, 2015
(http://dx.doi.org/10.1108/YC-12-2014-00495). For what it’s worth, such findings also reproduce for
adults, and not only in Western samples (e.g., Kim et al., 2018,
https://doi.org/10.1371/journal.pone.0198841).
Response:
At your suggestion, I used your references in the Introduction section, which was indeed, very
relevant. Please see below.
On the other hand, in his research, Eric Robinson concluded that interventions on raising
parental awareness regarding the weight status of their child needs to be aware of the possible
adverse outcomes on child’s mental health [28]. Literature showed that parental identification on
child overweight and obesity is associated with further gaining weight in youngsters [29] This
hypothesis is explained by the fact that heavy weigh children are teased and stigmatized inside family.
Also, a harsh parenting on weight-control, supported by criticisms and comments, has been
associated with emotional disorders in children [28].
Moreover, evidence showed that obesity is susceptible to mental health problems, because of
weight stigma [30]. Hence, a heavy child who experience stereotypical portraits, discriminations
(mistreatments, social devaluation), harsh language towards him, is likely to engage in disordered
eating [31]. Prior findings suggest that weight stigma is a predictor for future weight gain [32] which
can be explained by the fact that it increases the obesogenic stress process and triggers coping
mechanisms like overeating [33].

3. Labelling is also associated with fewer health behaviours and great levels of unhealthy
behaviours. In contrast, weight satisfaction (vs dissatisfaction), even in higher-weight individuals, is
linked to better mental and physical health and better long-term health outcomes, including reduced
incident of diabetes and other chronic conditions
(see e.g, Muennig et al., 2008,
doi:10.2105/AJPH.2007.114769; Blake et al., 2013, doi: 10.1155/2013/291371; Wirth et al., 2015,
http://dx.doi.org/10.1016/j.bodyim.2014.11.003; and 2014, doi:10.1037/hea0000058). For a review
of some of this literature, including the reasons why the obesity
 dieting  improved health paradigm
is not supported by the evidence, see a recent review by Hunger et al., 2020 (DOI:
10.1111/sipr.12062). This review is actually not exhaustive on the literature around labelling/weight
perception, and is not focussed on children’s health (although it discusses BMI surveillance, weight
report cards, labelling etc). I do not have time to search for other papers on weight misperception, but
the authors should consider themselves under a moral obligation to do so if they intend to publish this
work. Your one mention of this conundrum is in the Discussion – your ref 40 – says the same, correct
‘overweight’ perception is associated with greater weight loss intentions but not with healthy
behaviours.
Response
In the Introduction, we added additional information about labelling in children and we used
your suggested reference. Please see below.
“Individuals may experience, additionally, other forms of weight stigma, such as teasing,
bullying, and labelling which can have a significant impact on well-being, given the high sensitivity

of weight issues during adolescence [31]. The “too fat” label, especially when it comes from family,
has long-term detrimental effects on health. Considerable studies showed that labelling in childhood
led to distorted eating cognitions and behaviors tracked into adulthood and it was associated with
high BMI years later, regardless the initial weight [31,33]. “
4. In your discussion, you note as a possible explanation for parental misperceptions: “Last but
not least, parents’ concern is that their child is in good health, meaning being happy and active, rather
than relating to the body size per se [45,48].” This is absolutely how it should be.
Response
We reformulate this part of discussion. Please see below
Parents’ bias regarding children’s body weight could be explained by several factors. First, since
the prevalence of obesity in children increased lately, the view of the community regarding excess
weight has been desensitized and nowadays it is perceived as normal [63,64]. Because of lack of
knowledge, parents tend to estimate the children’s body weight through visual comparison with other
children, who subjectively are perceived as overweight. Second, parents might be reluctant to accept
that their child as overweight or obese, because of social pressures [64]. On the other hand, parents
misperception is rather an emotional evaluation, rather than a cognitive bias [65].
5. Similar issues to those associated with labelling/correct perception arise with parental input,
comments and food-control behaviours around children’s weight. These also tend to be associated
with more disordered eating, negative psychological outcomes, and weight gain – again, the authors
are encouraged to explore this literature before publishing this work – the suggestion that parental
intercession in child weight-control is a positive goal should be reconsidered. There is no doubt that
misperception can mean fewer dieting attempts and less parental effort to get children to lose weight
– that this is problematic is only assumed, based on equating weight with health. You say that
“recognition is a critical step in obesity prevention.” We do not have data that suggest telling children
they are overweight, or getting their parents to notice they are overweight, and do something about
this provides any form of long-term positive outcome
Response
In the Conclusion section section we added information about the importance of appropriate
communication and attitudes of parents regarding weight management of their children in order to
avoid parents ‘negative effects on children. Please see below.
Moreover, health care professionals should aim to raise parental awareness through
counselling. For example, the healthcare professional could use the BMI as a screening tool to
monitor the child’s weight status during periodical consultations, explain and reflect with parents on
risks of obesity and appropriate tools for communicating with their children, in order to avoid
inappropriate teasing and stigmatization or pressure to lose weight, which might have negative effect
on both nutrition and weight related behaviour of children as well as on their mental health and
wellbeing, as other studies also suggest (51,55). Also, physicians should support parents to
encourage a healthy lifestyle inside family.
6. Even more extensive is the well–documented negative impact of weight stigma on child and
adolescent health, including, again, weight gain, but also anxiety, depression, self-harm, and

suicidality independent of starting weight/BMI. Problematising fat bodies inevitably leads to this
negative perception and stigmatisation of fat children (and adults).
Response
In Introduction we included information about suicidality and other mental health disorders.
Please see below.
“Similar to suicidality, the prevalence of body image dissatisfaction is high in midchildhood
[43]. In his publication, Kline outlined that normal weight children who saw themselves heavier
reported more often suicidal thoughts than obese teens who perceived their weight accurately [44].
Beside suicidal ideation, body image dissatisfaction is comorbid with anxiety, depression, self-injury
and low self-esteem [45].”
7. Assuming the authors are willing to reframe their study, I also have a few minor comments
on the study itself and reporting of the results that I hope the authors find helpful. These are intended
to help improve clarity and do not reflect on the quality of the paper:
School is not a ‘demographic characteristic’ as such. Was this used in some way as a proxy
for socioeconomic status? Was SES measured and controlled for? It might be helpful to include a
summary of the sample demographics
Response
We have eliminated “school” from demographic characteristic because it had nothing to do
with socio-economic status.
In the Limitations part we mention that the socio-economic factors were not evaluated.
Desired weight: what does ‘none of the above’ refer to? Was this a ‘prefer not to answer’
option?
Response
In section 2.2 we clarified better that “none of the above” refers to a prefer- not to- answer
option.
Slimming tea and massage are not common forms of weight control behaviours in most
countries that I am aware of. This might be a local phenomenon? It might be worth expanding on
this a little – in particular, I can imagine that ‘slimming tea’ may be supposed to have some fat
burning compounds, but how is massage supposed to lead to weight loss?
Response
Several methods for losing weight are advertised and might be present to some extant among
different population groups, even if they are unhealthy or with no clear benefits, so we included these
methods which are advertised or adopted to some extent by some groups.
The descriptions of physical activity are not consistent. Defined as ‘at least 10 minutes’ but
‘available responses ranged from half an hour to more than 2 hours.’ Do you mean ‘actual responses’
rather than ‘available responses’? Please clarify

Response
We better explain now the assessment of physical activity. Please see below
“Physical activity (PA) performed in the last week was assessed through 2 items. The first
item referred to the frequency and the possible options ranged from Never to 7 days a week. The
second item assessed the duration of physical activity performed during those days. The average time
dedicated to PA was calculated by multiplying the frequency by the duration (in minutes) and dividing
by 7. “
The bullying question asks about bullying ‘because of their physical appearance or because
of their weight.’ These are not necessarily the same thing. Were these items asked separately or was
that a single item? Additionally, was this question about in-person bullying only, with the following
person asking about cyberbullying, or was the first question about ALL bullying and the second
question just one component of this? Please clarify.
Response
In section 2.2 we clarified better your suggestions. Please see below.
Bullying-As a single item, children were asked if they have been bullied by others (bad words,
laughter, shouting, threats towards them) in the last 3 months because of their physical appearance
or because of their weight. Responses rated to a 4-point scale (Never, Once, 2-3 times, More than 3
times).
Cyber-bulling-Separately from the previous question, children were asked if they have been
bullied on phone, e-mail or social media by others (insults, bad words, threats towards them) in the
last 3 months because of their physical appearance or because of their weight. Responses rated to
a 4-point scale (Never, Once, 2-3 times, More than 3 times).
If children have ever been excluded from the group (don’t talk, ignore, avoid them) in the last
3 months because of their physical appearance or because of their weight. Responses rated to a 4-
point scale (Never, Once, 2-3 times, More than 3 times).”
Children were asked if they had discussed their weight with their parents in the previous year.
How accurate is this likely to be? Especially in younger children. In addition to the issue of recall,
the time frame is also likely to pose problems to most children. Also, responses are likely
systematically biased, with children who have ‘problem’ weight, more likely to remember such a
family discussion.
Response
We acknowledge this limitation in the Limitation part. Please see below.
“This study is subject of several limitations. First, our sample included children from 4 urban
settings from North-West part of Romania, therefore, the generalization of the provided information
to the whole country is limited. Second, we did not assess the relationship between socio-economic
status and children’s weight. Third, data on bullying and physical activity were based on a selfreported questionnaire that could influence our findings due to memory bias. Perceptions of
children’s weight were assessed through questioning teenagers how they estimate their weight (too
big, too small, or normal), instead of using specific tools for this issue. Moreover, due to the crosssectional design of the study, we could not determine the longitudinal consequences of weight

behaviours on children’s BMI and also, could not explore the long-term parental effects on children’s
physical and mental wellbeing. Moving further, the preliminary results on bullying give impetus to
further research on the effects of weight stigma on children’s mental health and its relationship to
disordered eating. In the end, the sample of parents was modest and our data was based on their
statements. Further studies should focus more on parents’ motivation to participate and assess
differences in gender perceptions.”
In data analysis, what is meant by ‘underweight and thin’? Is this one category? Is this a
translation issue?
Response
We removed “thin” from text, due to a translation issue.
Bottom of page 4: You use the sentence, “Parents’ assessment regarding children’s body size
was similar. Their possible responses were dichotomized into two categories (Correct/Incorrect
estimation), whether their perception on child’s weight status matched or not with children’s actual
BMI.” This is a beautifully succinct description that could be use for both children and parents. The
preceding three paragraphs saying essentially the same thing are quite repetitive and confusing
Response
At your suggestion we used a succinct description for both children and parents. Please see
below.
There were used logistic regression analyses to examine the relationship between several
factors and higher BMI in children (coded as 0= Normal weight, 1=Excess weight) as well as
differences in misperceptions of children’s weight (for normal weight children we investigated two
subcategories: correct estimation vs incorrect underestimation and correct estimation vs incorrect
overestimation, while for excess weight children, we evaluated correct estimation vs incorrect
underestimation); correct estimation was considered when the perceived weight by children was in
the same category as the actual weight, while when the perceived weight was in a higher or lower
category than that of actual weight it was considered overestimation, respectively underestimation.
The independent variables considered were gender (girls=1, boys=0), age, personal estimation of
weight (only for BMI) and parental estimation regarding weight based on whether their perception
on child’s weight status matched or not with children’s actual BMI (Correct=1, Incorrect=0;.”),
children’s weight management (Attempts to lose weight coded as Yes=1, No=0 and Attempts to gain
weight coded as Yes=1, No=0), discussion between children and their parents on weight issues
(Yes=1, No=0,), discussion between parents and a healthcare professional regarding their child’s
weight (Yes=1, No=0), time spent on physical activities ( <1 hour=1, 1-2 hours=2, >2 hours=3) ,
the presence of bulling, cyber bulling and or being excluded from the group (Never=0, <3 times=1,
≥3 times=2 ).”
Logistic regression coding. 0 = NW, 1 = EW. What about underweight? Were they excluded
because of low numbers? Please clarify.
Response
We clarified in section 2.3 that underweight children were excluded because of low number.

In the table note for T1, you indicate that the asterisk marks that type of restricted eating
analysis was reported for those who indicated food restriction. Please insert the N either in the table
note or in the table itself. (Tables 1 and 2)
Response
We inserted N in the table for food restrictions and for physical activity as well. Please see
Table1.
Tables 2 and 3. I think it would be more helpful to see these side by side, rather than having
to go backwards and forwards between them. Could you please combine into a single table.
Response
We combined Tables 2 and 3 into one. Please check Table 2.
Just a query – purging (vomiting) in boys was almost double that in girls? Is that correct?
Response
Yes, we checked the results once again.
End of section 3.1, girls more concerned to discuss weight with family: I think it’s more likely
to be the other way round – parents are more concerned about discussing weight with female children
than male children – this is consistent with body ideal norms being enforced more strongly in girls,
greater body objectification and self-objectification in girls, and girls experiencing more weight
stigma and at lower BMI than boys. Maybe mention this in the discussion – this gender difference
indicates that parental weight talk is definitely more about social norms than about health, otherwise
it’d be the same for both.
Response
In Discussion section we added a possible explanation for gender differences. Please see
below.
“There were noticed gender differences, girls declaring they have spoken more often with
parents about weight, rather than boys did. The explanations could be that females received from
their family socio-cultural messages about how their weight should be, but also the fact that girls are
more interested and opened to discuss these issues with their parents, as a consequence of their
uncertainties, look for information or desire to share their concerns”
Section 3.2 heading is misleading. Suggest changing it to, “Characteristics of children in
different weight categories”
Response
We changed the heading to “Characteristics of children in different weight categories”.
Table 4 is really confusing to follow. There must be a better way to report these findings?
Smaller adjustments to Table 4 that might help:
- Remove the single bullet point for non-significant findings. The absence of the traditional
asterisks is indicative of lack of significance.

- BMI should be ‘BMI category’
-Also, table headings are confusing by inferring a reference category that is not necessary. Perhaps
just use Normal weight children ‘underestimators’ and ‘overestimaters,’ and Excess weight children
‘underestimaters.’
- Similarly, instead of using two rows and including a reference row ‘1’, you could just put it
in the variable name (e.g., “Female (vs. male)).
- I had several other comments/queries but they might be due to confusion because of the
double reference category (columns and rows) – fix that first and hopefully it’ll become clearer.
- Are the WL attempt figures in the correct columns? Maybe just simplify headings and then
double check all the numbers
Response
We followed your recommendations and changed inside table. Please check Table 3. We also
checked numbers once again.
Discussion, “girls tended to lose weight more frequently than boys” – be careful here – correct
to “girls attempted weight loss more frequently than boys.” The most common outcome of weight
loss attempts is actually weight gain.
Response
Thank you for your mention. We corrected to “girls attempted weight loss more frequently
than boys”.
% trying to gain weight: Weight gain attempts were more frequent in boys – is it possible this
was about gaining muscle rather than fat? I can’t imagine any reason why a child would want to gain
fat tissue because others in their family or among their peers were fatter. This needs to be better
explained.
Response
Thank you for your suggestion. We explained in Discussion section differences on gender for
gaining weight. Please see below.
“Nevertheless, almost half of underweight children tried to gain weight in the previous year,
whereas 65.1 % of fat children self-reported attempts to lose wight the year before, which might
suggest that the desired weight is associated to a healthy weight. On gender, underweight boys were
more concerned to gain weight and overweight girls to lose weight. A likely explanation would be
that underweight males tried to gain muscle mass, whereas excess weight females tried to achieve a
slim body. “
The manuscript would also benefit from a proofread by a native English speaker.
Response
The English was checked and improved.

Reviewer 2 Report

The introduction is the part of the scientific article in which the author informs what was researched and why the investigation was carried out. I suggest that authors try to meet this demand of Scientific Writing. IS GOOD SECTION.

But, my suggested for adequation is here: This aim of this study was to assess the actual weight, perceived weight and desired weight of Romanian school children, looking to both opinions and practices of children and their parents.

My suggested: This aim of this study was to assess the actual weight, perceived weight and desired weight of Romanian school children.

Some questions help in writing. What is the study about? Why was it done? Why should it be published? I've missed those answers throughout this Introduction Section.

It also seeks to show that the research is based on solid foundations. Thus, in the introduction, a link is made with the relevant literature. What was known about the subject at the beginning of the investigation? What was not known about the matter and motivated the investigation? Answering these questions involves a process of choosing the works to cite. in the section I got this. Good writing and empirically based on solid literature

An outcome is suggested at the end of the Introduction Section, before the Objective Section: among the criteria used to choose them are relevance, accessibility and timeliness.

The first three parts of the body of an original article – introduction, method and results – are essential for the composition of this Discussion Section.

In this text, the discussion section is essential for the closing of the manuscript, as it completes the structure of the article.

The discussion is the place of the article that houses the comments on the meaning of the results, the comparison with other research findings and the author's position on the subject. A discussion without a coherent structure is displeasing, hence the convenience of organizing themes into topics.

Thus, my suggestion to the authors is to provide the separation of this Discussion Section into topics, as follows:

- Highlight relevant and original findings in the Results section; Critical evaluation of the research itself: limitations and positive aspects;

- Critical comparison with the relevant literature ) insert here the findings of the articles registered during the article construction process; Interpretation of research findings;

- and in the last paragraph, insert the Conclusion Section, which may be accompanied by generalization, implications, perspectives, recommendations in the field of the object of the epidemiological study proposed by the authors.

Each of the topics must inform about a facet of the discussion and as a whole provide the subsidies to judge the adequacy of the arguments, the conclusion and the entire text.

A convenient way to start the Discussion is to highlight, with a few words, the most important findings or new knowledge revealed by the research.

After this initial part, the method used is commented, so that the author informs how valid the research seems to him. It is considered good practice for the author himself to point out the shortcomings rather than deliberately omitting them, hoping that they will go unnoticed.

Important limitations not mentioned in the text diminish the credibility of the investigation. The limitations that can substantially influence the results and change the conclusions of the investigation deserve to be pointed out.

These limitations are related to the type of design used or the details of the investigation itself. Positive aspects are also commented on, including the measures taken to neutralize the limitations, to circumvent them or estimate their influence on the results. Authors are suggested to highlight these points in the Discussion section.

The relationship of the research findings with the relevant knowledge available at the time of writing the manuscript.

The interpretation of comparisons between studies is problematic in the presence of methodological differences. This issue will be resolved in the case of providing a comparison between the studies included in the records and resulting from the search for an empirical basis and the presentation of the novelty of each one of them. Example: It only makes sense to compare frequencies when produced in a similar way. If, in one research, the data are obtained through an interview and, in another, through the verification of medical records, the differences found may only reflect the form of data collection.

Many other factors explain the variation in results achieved by different investigations, among which are the types of design, the scenarios in which the research is carried out, the classification criteria to include or exclude patients from the sample, the definitions of variables, the characteristics of the studied groups, the content of the interventions and the sample size.

Thus, the specificities and quality of the works, their limitations and their positive aspects are taken into account in the reference to other articles present in the discussion. The reader will benefit from becoming familiar with systematic reviews, especially in the aspects concerning the collection, evaluation and classification of the quality of articles. The comparison of data between methodologically homogeneous studies allows concluding, with greater conviction, whether the results in the literature agree or not with those of the research reported. THIS IS A GOOD DISCUSSION OF A SCIENTIFIC ARTICLE. The authors failed to accomplish this task.

When the results point in the same direction, the discussion is simpler to be conducted, however this does not seem to me to be the case in this Section described here. If there are marked discrepancies between the findings, these discrepancies are recorded and commented on in an attempt to clarify the possible reasons for the differences. Impartiality is a highly appreciated feature of researchers. It manifests itself in many ways, one of which is to include, in the discussion, reports that do not coincide with the results of the investigation itself. The interpretation of research results implies the search for a plausible explanation for the findings.

To this end, other explanations are excluded before deciding on the most likely one. We will have more conviction in the conclusion if bias and chance have been eliminated as an explanation for the findings. Even after bias and chance are removed, there may be more than one possible explanation.

Every scientific investigation report needs a conclusion. The conclusion is the position of the author of the study, consistent with its objectives and the report itself. This is because there is a conclusion that, necessarily, is included in the study summary. As this is situated earlier, at the beginning of the article, the author will assess whether it is worth repeating the conclusion in the discussion. I suggest reviewing this structure for manuscript submission to a journal in the field of knowledge.

It is worth noting that in the Conclusion Section there is still room for speculation and implications. For example, signaling the direction of future efforts and recommendations for further research. The discussion is usually a difficult section to prepare. It is the one in which the beginner gets the most complicated in his writing and, commonly, elaborates a long and confusing text. Furthermore, as a suggestion for size, this section should not exceed a third of the article. As a quality of this text, it seemed to me that there was an association between beginners and experienced researchers in scientific communication, which was positively reflected in the argumentation of the writing of this Discussion Section.

Additional comments

Experienced writers organize the introduction to pique the reader's interest and keep them reading. Those who write want to be read, quoted and hope that their information will be useful to the community. In order to please readers and scientific editors, the text must have certain characteristics, including conciseness, clarity, accuracy, logical sequence and elegance. JHGD appreciates short introductions, but with sufficient and adequate information. To get text with such attributes, it is good to remember the three rules for writing well: proofread, proofread, proofread. Those are the suggestions!

Also, in the conclusion section, the authors describe a long text and do not respond to the objectives.

“Our study revealed misperceptions on child’s weight among them and their parents, which may influence negatively their current healthy lifestyle with risk of delayed intervention on body weight management and development of future health complications derived from obesity. Further efforts should be directed towards weight related issues in which both children and parents should be involved actively. These initiatives need to be included into a multidimensional approach in which educational messages should be delivered not individually, but through schools, healthcare professionals as well as community working closely together. Implementation in schools of educational programs as well as designing appropriate education and information actions and facilitating access to educational resources could help pupils understand the principles for a healthy diet, dispel the myth of an ideal body that children want to achieve, prevent and reduce the trend of childhood obesity. As recognition is a critical step in obesity prevention, pediatricians should aim to raise parental awareness through counselling. For example, the healthcare professional could use the BMI as a screening tool to monitor the child’s weight status during periodical consultations, explain and reflect with parents on risks of obesity. Also, physicians should support parents to encourage a healthy lifestyle inside family. Rather than focusing on body weight management, interventions should promote means to increase the overall quality of life.

The text contained in the CONCLUSION section is part of the Discussion section. Thus, it must be inserted in the final section.

The authors should respond to the research objective within the conclusion item.

Author Response

Response letter

We thank you for your quick response and we would like to thank the reviewers for the through feedback. We have adjusted our article according to their suggestions, and their work was invaluable in revising our work.

We have added our answers to every question received from the reviewers in this letter below. We have also highlighted the changes in our article as well as in this letter, under every question placed by the reviewers.

Reviewer 2

  1. The introduction is the part of the scientific article in which the author informs what was researched and why the investigation was carried out. I suggest that authors try to meet this demand of Scientific Writing. IS GOOD SECTION. But, my suggested for adequation is here: This aim of this study was to assess the actual weight, perceived weight and desired weight of Romanian school children, looking to both opinions and practices of children and their parents. My suggested: This aim of this study was to assess the actual weight, perceived weight and desired weight of Romanian school children.

Response

Thank you for suggestion. We changed it to “The aim of this study was to assess the actual weight, perceived weight and desired weight of Romanian school children”

  1. Some questions help in writing. What is the study about? Why was it done? Why should it be published? I've missed those answers throughout this Introduction Section. It also seeks to show that the research is based on solid foundations. Thus, in the introduction, a link is made with the relevant literature. What was known about the subject at the beginning of the investigation? What was not known about the matter and motivated the investigation? Answering these questions involves a process of choosing the works to cite. in the section I got this. Good writing and empirically based on solid literature

Response

We added in Introduction section information regarding the current situation in Romania. Please see below.

In Romania, national information on the prevalence of overweight and obesity among children is limited. A review of the studies published between 2006-2015 showed that 25% of youth aged 6-19 years old were overweight and obese [49]. A recent study showed that prevalence of overweight and obesity among children and adolescents aged 7-18 years old from western part of the country was of 30% and it was highest among pre-pubertal children and boys [50].

Starting with the 90’s, Romania has undergone socio-economic changes that might influence the trend of obesity.  To our knowledge, few studies on the Romanian population assessed the association between children’s BMI and weight control behaviors. Also, prior research on Romanian children and parents’ perception regarding body weight is scarce and there is no previous evidence about the relationship between children’s weight status and vulnerability to bullying in Romania.

Hence, the present study has four objectives. First, it evaluated the correspondence between anthropometric assessments of Romanian children (actual weight) and their perceptions of their weight (children’s perceived weight) as well as their parents’ perception of their children’s weight (perceived weight by parents), but also their concern for body weight management among children (desired weight). Second, the study assessed weight management and lifestyle related behaviors, as well as the prevalence of bullying among participating children and communication between children and parents and between parents and healthcare professionals regarding weight. Third, the factors associated with overweight and obesity were evaluated. Last, but not least, we aimed to identify the factors which influence the correct estimation of their weight by children.”

  1. An outcome is suggested at the end of the Introduction Section, before the Objective Section: among the criteria used to choose them are relevance, accessibility and timeliness.

The first three parts of the body of an original article – introduction, method and results – are essential for the composition of this Discussion Section.

In this text, the discussion section is essential for the closing of the manuscript, as it completes the structure of the article.

The discussion is the place of the article that houses the comments on the meaning of the results, the comparison with other research findings and the author's position on the subject. A discussion without a coherent structure is displeasing, hence the convenience of organizing themes into topics.

Thus, my suggestion to the authors is to provide the separation of this Discussion Section into topics, as follows:

- Highlight relevant and original findings in the Results section; Critical evaluation of the research itself: limitations and positive aspects;

- Critical comparison with the relevant literature ) insert here the findings of the articles registered during the article construction process; Interpretation of research findings;

- and in the last paragraph, insert the Conclusion Section, which may be accompanied by generalization, implications, perspectives, recommendations in the field of the object of the epidemiological study proposed by the authors.

Each of the topics must inform about a facet of the discussion and as a whole provide the subsidies to judge the adequacy of the arguments, the conclusion and the entire text.

A convenient way to start the Discussion is to highlight, with a few words, the most important findings or new knowledge revealed by the research.

After this initial part, the method used is commented, so that the author informs how valid the research seems to him. It is considered good practice for the author himself to point out the shortcomings rather than deliberately omitting them, hoping that they will go unnoticed.

Important limitations not mentioned in the text diminish the credibility of the investigation. The limitations that can substantially influence the results and change the conclusions of the investigation deserve to be pointed out.

These limitations are related to the type of design used or the details of the investigation itself. Positive aspects are also commented on, including the measures taken to neutralize the limitations, to circumvent them or estimate their influence on the results. Authors are suggested to highlight these points in the Discussion section.

The relationship of the research findings with the relevant knowledge available at the time of writing the manuscript.

The interpretation of comparisons between studies is problematic in the presence of methodological differences. This issue will be resolved in the case of providing a comparison between the studies included in the records and resulting from the search for an empirical basis and the presentation of the novelty of each one of them. Example: It only makes sense to compare frequencies when produced in a similar way. If, in one research, the data are obtained through an interview and, in another, through the verification of medical records, the differences found may only reflect the form of data collection.

Many other factors explain the variation in results achieved by different investigations, among which are the types of design, the scenarios in which the research is carried out, the classification criteria to include or exclude patients from the sample, the definitions of variables, the characteristics of the studied groups, the content of the interventions and the sample size.

Thus, the specificities and quality of the works, their limitations and their positive aspects are taken into account in the reference to other articles present in the discussion. The reader will benefit from becoming familiar with systematic reviews, especially in the aspects concerning the collection, evaluation and classification of the quality of articles. The comparison of data between methodologically homogeneous studies allows concluding, with greater conviction, whether the results in the literature agree or not with those of the research reported. THIS IS A GOOD DISCUSSION OF A SCIENTIFIC ARTICLE. The authors failed to accomplish this task.

When the results point in the same direction, the discussion is simpler to be conducted, however this does not seem to me to be the case in this Section described here. If there are marked discrepancies between the findings, these discrepancies are recorded and commented on in an attempt to clarify the possible reasons for the differences. Impartiality is a highly appreciated feature of researchers. It manifests itself in many ways, one of which is to include, in the discussion, reports that do not coincide with the results of the investigation itself. The interpretation of research results implies the search for a plausible explanation for the findings.

To this end, other explanations are excluded before deciding on the most likely one. We will have more conviction in the conclusion if bias and chance have been eliminated as an explanation for the findings. Even after bias and chance are removed, there may be more than one possible explanation.

Every scientific investigation report needs a conclusion. The conclusion is the position of the author of the study, consistent with its objectives and the report itself. This is because there is a conclusion that, necessarily, is included in the study summary. As this is situated earlier, at the beginning of the article, the author will assess whether it is worth repeating the conclusion in the discussion. I suggest reviewing this structure for manuscript submission to a journal in the field of knowledge.

It is worth noting that in the Conclusion Section there is still room for speculation and implications. For example, signaling the direction of future efforts and recommendations for further research. The discussion is usually a difficult section to prepare. It is the one in which the beginner gets the most complicated in his writing and, commonly, elaborates a long and confusing text. Furthermore, as a suggestion for size, this section should not exceed a third of the article. As a quality of this text, it seemed to me that there was an association between beginners and experienced researchers in scientific communication, which was positively reflected in the argumentation of the writing of this Discussion Section.

Response

Thank you for your suggestions. In Introduction we reformulated the objectives of our study. We also added additional resources for several issues, such as risk of weight gain in overweight children, weight stigma, the risk of suicidal thoughts and other mental disorders in children, the negative effects of parenting, the risk of metabolic syndrome due to weight dissatisfaction.

We also highlighted the relevant findings in Results section. We improved Discussion section, by commenting the results of previous research and highlighted our novelty findings. Moreover, we correlated our results to prior research. We also added future directions in the field, and we outlined the limitations for our research. Please see below.

“On the other hand, in his research, Eric Robinson concluded that interventions on raising parental awareness regarding the weight status of their child needs to be aware of the possible adverse outcomes on child’s mental health [28]. Literature showed that parental identification on child overweight and obesity is associated with further gaining weight in youngsters [29] This hypothesis is explained by the fact that heavy weigh children are teased and stigmatized inside family. Also, a harsh parenting on weight-control, supported by criticisms and comments, has been associated with emotional disorders in children [28]. “

“Moreover, evidence showed that obesity is susceptible to mental health problems, because of weight stigma [30]. Hence, a heavy child who experience stereotypical portraits, discriminations (mistreatments, social devaluation), harsh language towards him, is likely to engage in disordered eating [31]. Prior findings suggest that weight stigma is a predictor for future weight gain [32] which can be explained by the fact that it increases the obesogenic stress process and triggers coping mechanisms like overeating [33].  Individuals may experience, additionally, other forms of weight stigma, such as teasing, bullying, and labelling which can have a significant impact on well-being, given the high sensitivity of weight issues during adolescence [31]. The “too fat” label, especially when it comes from family, has long-term detrimental effects on health. Considerable studies showed that labelling in childhood led to distorted eating cognitions and behaviors tracked into adulthood and it was associated with high body mass index (BMI) years later, regardless the initial weight [31,33]. “

“Similar to suicidality, the prevalence of body image dissatisfaction is high in midchildhood [43]. In his publication, Kline outlined that normal weight children who saw themselves heavier reported more often suicidal thoughts than obese teens who perceived their weight accurately [44].  Beside suicidal ideation, body image dissatisfaction is comorbid with anxiety, depression, self-injury and low self-esteem [45].”

“On the other hand, evidence showed that chronic weight dissatisfaction may affect the physical health as well. High weight discrepancy was linked to poor dietary, lack of physical activity, tobacco use and alcohol consumption which consequently led to an increased likelihood of type 2 diabetes, hypertension, and metabolic syndrome later in life [46,47]. On the contrary, people who declared satisfied with their current weight, regardless BMI, were more likely to adopt healthy lifestyles and had better long-term health outcomes [48]. “

“The aim of this study was to examine the child-parent dyad related to the children’s perceived and desired weight in contrast to the children’s actual weight. Also, within this study, we explored the relationships between under- and over-estimation in normal and heavy weight children, looking to both weight-related experiences and behaviors of them and their parents. To our knowledge, this study would be the second research conducted in Romania that evaluated these issues [27]. Compared to the previous study which was carried out in 2013 on a group of 344 students from a city of Romania, our study was conducted in 2019 and included a larger sample of children from several cities of Romania which allowed separate analyzes for girls and boys. Moreover, it included additional topics such as physical activity, hours of sleep and types of bullying that haven’t been assessed before, and we examined their relationship to the actual weight and perceived weight.”

On the other hand, unlike the previous study, 20% of underweight children reported attempts to lose weight in the previous year as well. This finding could also be attributed to body weight dissatisfaction.  Body image distortion includes negative beliefs concerning body shape and appearance (cognitions), misperception (incorrect evaluation of individual own’s size, shape, and weight), an affective compound and a behavioral impairment (such as body checking, unhealthy weight-related behaviors, dieting, isolation) which are core factors for the development of mental health problems, including disorder eating or even body dysmorphic disorder [55]

Parents’ bias regarding children’s body weight could be explained by several factors. Firstly, since the prevalence of obesity in children increased lately, the view of the community regarding excess weight has been desensitized and nowadays it is perceived as normal [63,64]. Because of lack of knowledge, parents tend to estimate the children’s body weight through visual comparison with other children, who subjectively are perceived as overweight. Second, parents might be reluctant to accept that their child as overweight or obese, because of social pressures [64]. On the other hand, parents misperception is rather an emotional evaluation, rather than a cognitive bias [65]. “

There were noticed gender differences, girls declaring they have spoken more often with parents about weight, rather than boys did. The explanations could be that females received from their family socio-cultural messages about how their weight should be, but also the fact that girls are more interested and opened to discuss these issues with their parents, as a consequence of their uncertainties, look for information or desire to share their concerns.

“This study is subject of several limitations. First, our sample was recruited from a single region, therefore, the generalization of the provided information to the whole country is limited. Second, we did not assess the relationship between socio-economic status and children’s weight. Third, although our study was conducted in urban setting, it also included children from rural areas, in the vicinity of the participating schools. Forth, data on bullying and physical activity were based on a self-reported questionnaire that could influence our findings due to memory bias. Perceptions of children’s weight were assessed through questioning teenagers how they estimate their weight (too big, too small, or normal), instead of using specific tools for this issue. Moreover, due to the cross-sectional design of the study, we could not determine the longitudinal consequences of weight behaviors on children’s BMI and also, could not explore the long-term parental effects on children’s physical and mental wellbeing. Moving further, the preliminary results on bullying give impetus to further research on the effects of weight stigma on children’s mental health and its relationship to disordered eating.  In the end, the sample of parents was mixed but also modest and our data was based on their statements. Further studies should focus more on parents’ motivation to participate and assess differences in gender perceptions.”

“Concentrated efforts should be directed towards weight related issues in which both children and parents should be involved actively. These initiatives need to be included into a multidimensional approach in which educational messages should be delivered not individually, but through schools, healthcare professionals as well as community working closely together. Implementation in schools of educational programs as well as designing appropriate education and information actions and facilitating access to educational resources could help pupils understand the principles for a healthy diet, dispel the myth of an ideal body that children want to achieve, prevent and reduce the trend of childhood obesity. Moreover, pediatricians should aim to raise parental awareness through counselling. For example, the healthcare professional could use the BMI as a screening tool to monitor the child’s weight status during periodical consultations, explain and reflect with parents on risks of obesity. Also, physicians should support parents to encourage a healthy lifestyle inside family.”

  1. Additional comments

Experienced writers organize the introduction to pique the reader's interest and keep them reading. Those who write want to be read, quoted and hope that their information will be useful to the community. In order to please readers and scientific editors, the text must have certain characteristics, including conciseness, clarity, accuracy, logical sequence and elegance. JHGD appreciates short introductions, but with sufficient and adequate information. To get text with such attributes, it is good to remember the three rules for writing well: proofread, proofread, proofread. Those are the suggestions!

Also, in the conclusion section, the authors describe a long text and do not respond to the objectives. “Our study revealed misperceptions on child’s weight among them and their parents, which may influence negatively their current healthy lifestyle with risk of delayed intervention on body weight management and development of future health complications derived from obesity. Further efforts should be directed towards weight related issues in which both children and parents should be involved actively. These initiatives need to be included into a multidimensional approach in which educational messages should be delivered not individually, but through schools, healthcare professionals as well as community working closely together. Implementation in schools of educational programs as well as designing appropriate education and information actions and facilitating access to educational resources could help pupils understand the principles for a healthy diet, dispel the myth of an ideal body that children want to achieve, prevent and reduce the trend of childhood obesity. As recognition is a critical step in obesity prevention, pediatricians should aim to raise parental awareness through counselling. For example, the healthcare professional could use the BMI as a screening tool to monitor the child’s weight status during periodical consultations, explain and reflect with parents on risks of obesity. Also, physicians should support parents to encourage a healthy lifestyle inside family. Rather than focusing on body weight management, interventions should promote means to increase the overall quality of life.  The text contained in the CONCLUSION section is part of the Discussion section. Thus, it must be inserted in the final section. (lines 507-520)

The authors should respond to the research objective within the conclusion item.

Response

We reformulated the objectives. Please see below.

“Hence, the present study has four objectives. First, it evaluated the correspondence between anthropometric assessments of Romanian children (actual weight) and their perceptions of their weight (children’s perceived weight) as well as their parents’ perception of their children’s weight (perceived weight by parents), but also their concern for body weight management among children (desired weight). Second, the study assessed weight management and lifestyle related behaviors, as well as the prevalence of bullying among participating children and communication between children and parents and between parents and healthcare professionals regarding weight. Third, the factors associated with overweight and obesity were evaluated. Last, but not least, we aimed to identify the factors which influence the correct estimation of their weight by children.”

We also included in the Conclusions section some data regarding the main findings, as well as the recommendations for future health promotion and research activities, based on the obtained results and in comparison with other studies. Please see below.

“The results show that 61.0% of pupils had normal weight, 7.4% were underweight and 31.6 % were overweight and obese. 66.7% of normal weight children, 56.5% of overweight children and 40% of underweight children perceived their weight accurately. Regarding parents, majority correctly appreciated the weight of their normal weight children and only a third the body weight of the underweight and overweight children. Factors such as body mass index, gender, weight related behaviors, parents’ estimation about their children’ weight, discussions inside family on weight topics and bullying were associated with misperceptions. The results have implications for future health promotion activities and research.

Concentrated efforts should be directed towards weight related issues in which both children and parents should be involved actively. These initiatives need to be included into a multidimensional approach with children, parents, schoolteachers, healthcare professionals as well as community working closely together. Implementation in schools of educational programs as well as designing appropriate education and information actions and facilitating access to educational resources could help pupils understand the principles for a healthy diet, dispel the myth of an ideal body that children want to achieve, prevent and reduce the trend of childhood obesity. On the other side, the results underline the importance of embracing these issues in health education curricula for children, the importance of prevention and decreasing of bullying being relevant also in relationship with physical and emotional well being of children, independent of their weight.

 Moreover, health care professionals should aim to raise parental awareness through counseling. For example, the healthcare professional could use the BMI as a screening tool to monitor the child’s weight status during periodical consultations, explain and reflect with parents on risks of obesity and appropriate tools for communicating with their children, in order to avoid inappropriate teasing and stigmatization or pressure to lose weight, which might have negative effect on both nutrition and weight related behavior of children as well as on their mental health and wellbeing, as other studies also suggest (51,55). Also, physicians should support parents to encourage a healthy lifestyle inside family.

Future research should pay attention to the development and evaluation of educational programs and measures for promoting appropriate weight, healthy lifestyle and a good quality of life among Romanian children.”

Reviewer 3 Report

The article entitled "Evaluation of the actual weight, perceived weight and desired weight of Romanian school children - views and practices of children and their parents" is very interesting.
The scientific research was conducted correctly and the results are adequately presented.
my question is about calculating the sample size. how was it calculated? Is the sample representative of the Romanian school population of the same age group?

Author Response

Response letter

We thank you for your quick response and we would like to thank the reviewers for the through feedback. We have adjusted our article according to their suggestions, and their work was invaluable in revising our work.

We have added our answers to every question received from the reviewers in this letter below. We have also highlighted the changes in our article as well as in this letter, under every question placed by the reviewers.

Reviewer 3

  1. The article entitled "Evaluation of the actual weight, perceived weight and desired weight of Romanian school children - views and practices of children and their parents" is very interesting.
    The scientific research was conducted correctly and the results are adequately presented.
    my question is about calculating the sample size. how was it calculated? Is the sample representative of the Romanian school population of the same age group?

Response

 We better explain now the recruitment process of the participants, the study being the first part of an educational program. Moreover, we mention in the limitations part the fact that the data could not be generalize at national level. Please see below.

“The study was approved by Ethics Commission of „Iuliu Hatieganu” Medicine and Pharmacy University, Cluj-Napoca, Romania (134/6.05.2019). It was implemented in 4 cities of 2 counties from North-West part of Romania (Cluj and Alba). In each county the study was implemented in two cities- the city capital of the county ( Cluj-Napoca for Cluj county, respectively Alba Iulia for Alba county) as well as other cities (Campia Turzii and Campeni respectively). School principals of 8 schools from the 4 cities were contacted and were informed about the objectives of the study and its characteristics and were asked if they are willing to allow the implementation of the study in the school; out of these, 7 school principals accepted to participate with their schools in the project and provided the number of the 5th to  8th grade classes that could participate. The parents of the children from these schools were contacted through letters, informing them about the study and informed consent was obtained from them regarding the children’s participation.

The study involved several phases- an evaluation of nutrition and weight management related opinions and behaviours of pupils (T1) followed by implementation of school based educational activities for promotion of healthy nutrition and active lifestyle. The present paper is based on data collected during T1.”

Limitations

“This study is subject of several limitations. First, our sample included children from 4 urban settings from North-West part of Romania, therefore, the generalization of the provided information to the whole country is limited”.